# Initiation of chromosome replication controls both division and replication cycles in *E. coli* through a double-adder mechanism

**Guillaume Witz[1,2†]\*, Erik van Nimwegen[1,2], Thomas Julou[1,2]**

[1]Biozentrum, University of Basel, Basel, Switzerland; [2]Swiss Institute of Bioinformatics, University of Bern, Bern, Switzerland

**Abstract** Living cells proliferate by completing and coordinating two cycles, a division cycle controlling cell size and a DNA replication cycle controlling the number of chromosomal copies. It remains unclear how bacteria such as *Escherichia coli* tightly coordinate those two cycles across a wide range of growth conditions. Here, we used time-lapse microscopy in combination with microfluidics to measure growth, division and replication in single *E. coli* cells in both slow and fast growth conditions. To compare different phenomenological cell cycle models, we introduce a statistical framework assessing their ability to capture the correlation structure observed in the data. In combination with stochastic simulations, our data indicate that the cell cycle is driven from one initiation event to the next rather than from birth to division and is controlled by two adder mechanisms: the added volume since the last initiation event determines the timing of both the next division and replication initiation events.

**\*For correspondence:**
guillaume.witz@math.unibe.ch

**Present address:** [†]Mathematical Institute, University of Bern, Bern, Switzerland

**Competing interests:** The authors declare that no competing interests exist.

## Introduction

Across all domains of life, cell proliferation requires that the chromosome replication and cell division cycles are coordinated to ensure that every new cell receives one copy of the genetic material. While in eukaryotes this coordination is implemented by a dedicated regulatory system in which genome replication and division occur in well-separated stages, no such system has been found in most bacteria. This suggests that the molecular events that control replication initiation and division might be coordinated more directly in bacteria, through molecular interactions that are yet to be elucidated. The contrast between this efficient coordination and the apparent absence of a dedicated regulatory system is particularly remarkable since most bacteria feature a unique replication origin which imposes that multiple rounds of replication occur concurrently in fast growth conditions. For example, in the specific case of *E. coli* that we study here, it has long been known that growth rate, cell size, and replication initiation are coordinated such that the average number of replication origins per unit of cellular volume is approximately constant across conditions (*Donachie, 1968*) or that cellular volume grows approximately exponentially with growth rate (*Taheri-Araghi et al., 2017*). Although several models have been proposed over the last decades to explain such observations (for a review and a historical perspective see for example *Willis and Huang, 2017*), so far direct validation of these models has been lacking, due to a large extent to the lack of quantitative measurements of cell cycles parameters in large samples with single-cell resolution.

Thanks to techniques such as microfluidics in combination with time-lapse microscopy, it has recently become possible to perform long-term observation of growth and division in single bacteria. By systematically quantifying how cell cycle variables such as size at birth, size at division, division time, and growth rate vary across single cells, insights can be gained about the mechanism of cell

cycle control. Several recent studies have focused on understanding the regulation of cell size, resulting in the discovery that *E. coli* cells maintain a constant average size by following an adder strategy: instead of attempting to reach a certain size at division (*i.e.* a sizer mechanism) or to grow for a given time (*i.e.* a timer mechanism), it was found that cells add a constant length $dL$ to their birth length $L_b$ before dividing (*Amir, 2014*; *Campos et al., 2014*; *Taheri-Araghi et al., 2017*). In particular, while the cell size at division and the division time correlate with other variables such the cell size at birth and growth rate, the added length $dL$ fluctuates independently of birth size and growth rate. A remarkable feature of the adder model is its capacity to efficiently dampen large cell size fluctuations caused by the intrinsically noisy regulation, without the need for any fail-safe mechanism. This efficient strategy has been shown to be shared by various bacterial species as well as by archea (*Eun et al., 2018*) and even some eukaryotes such as budding yeast (*Soifer et al., 2016*).

Here, we focus on how the control of replication initiation is coordinated with cell size control in *E. coli*, a question that has attracted attention for a long time (*Helmstetter, 1974*; *Pierucci, 1978*; *Koppes and Nanninga, 1980*). Several models have been proposed to explain how the adder behavior at the level of cell size might arise from a coordinated control of replication and division. Broadly, most models assume that the accumulation of a molecular trigger, usually assumed to be DnaA, leads to replication initiation, which in turn controls the corresponding future division event (*Campos et al., 2014*; *Ho and Amir, 2015*; *Wallden et al., 2016*). Subtle variations in how the initiation trigger accumulates and how the initiation to division period is set in each model imply distinct molecular mechanisms, and thus fundamentally different cell cycle regulations. Specifically, most models assume that initiation is triggered either when a cell reaches a critical absolute volume (initiation size, see for example *Wallden et al., 2016*) or alternatively when it has accumulated a critical volume since the last initiation event (see e.g. *Ho and Amir, 2015*). In order to explain the coordination between cell cycle events, division is often assumed to be set by a timer starting at replication initiation, but recent studies have also proposed that the two cycles might be independently regulated (*Micali et al., 2018a*; *Si et al., 2019*). Finally, it is often assumed that the regulation strategy could be different at slow and fast growth where different constraints occur.

We use an integrated microfluidics and time-lapse microscopy approach to quantitatively characterize growth, division, and replication in parallel across many lineages of single *E. coli* cells, both in slow and fast growth conditions. We show that insights about the underlying control mechanisms can be gained by systematically studying the structure of correlations between these different variables. Our single-cell observations are inconsistent with several previously proposed models including models that assume replication is initiated at a critical absolute cell volume and models that assume division is set by a timer that starts at replication initiation. Instead, the most parsimonious model consistent with our data is a double-adder model in which the cell cycle commences at initiation of replication and both the subsequent division and the next initiation of replication are controlled by the added volume. We show that this model is most consistent with the correlation structure of the fluctuations in the data and, through simulations, we show that this model accurately reproduces several non-trivial observables including the previously observed adder behavior for cell size control, the distribution of cell sizes at birth, and the distribution of the number of origins per cell at birth. Moreover, the same model best describes the data both at slow and fast growth rates. As far as we are aware, no other proposed model at the same level of parsimony can account for the full set of observations we present here.

## Results

To test possible models for the coordination of replication and division in *E. coli* we decided to systematically quantify growth, replication initiation, and division across thousands of single *E. coli* cell cycles, across multiple generations, and in various growth conditions. To achieve this, cells were grown in a Mother Machine type microfluidic device (*Wang et al., 2010*) and imaged by time-lapse microscopy. We used M9 minimal media supplemented with glycerol, glucose or glucose and eight amino acids, resulting in doubling times of 89, 53 and 41 min, respectively. The cell growth and division cycles were monitored by measuring single-cell growth curves obtained through segmentation and tracking of cells in phase contrast images using the MoMA software (*Kaiser et al., 2018*). The replication cycle was monitored by detecting initiation as the duplication of an *oriC* proximal FROS tagged *locus* imaged by fluorescence microscopy (*Figure 1A*). This of course only offers a proxy for

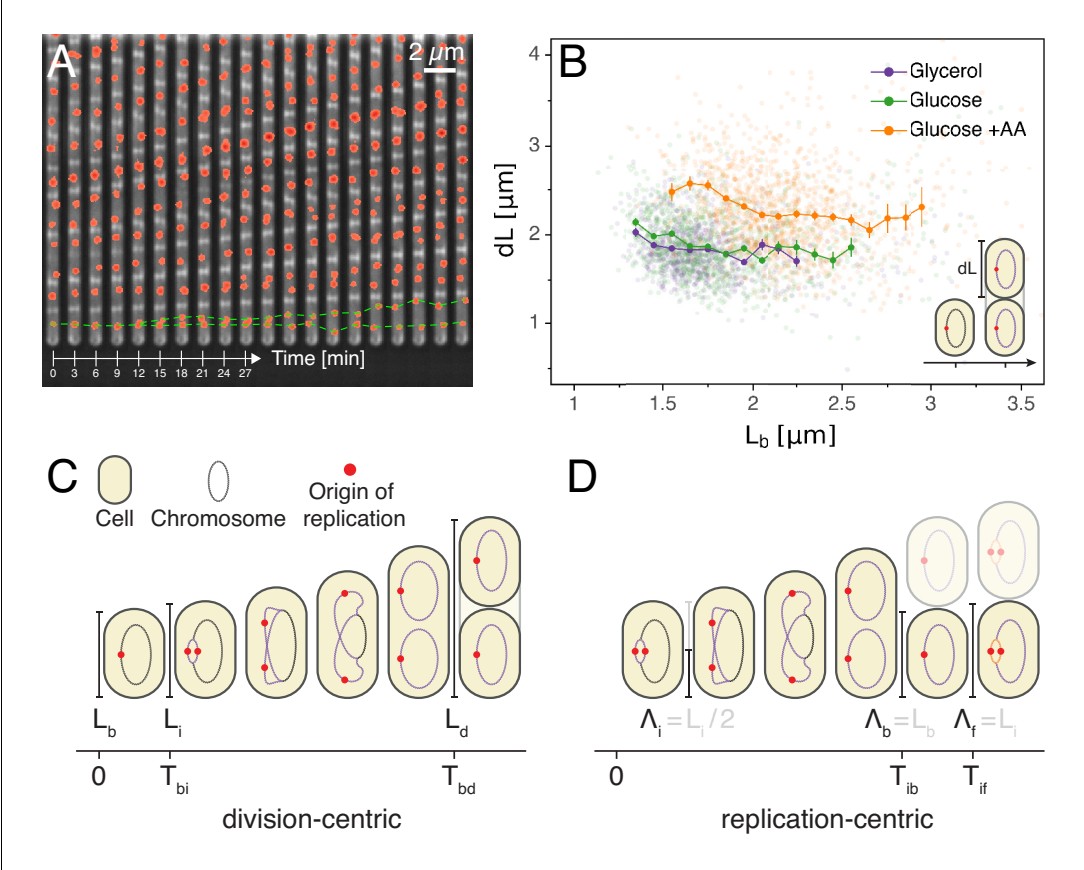

**Figure 1.** Experimental approach and analysis framework. (**A**) Time-lapse of *E. coli* cells growing in a single microfluidic channel. The fluorescence signal from FROS labeling is visible as red spots in each cell. The green dotted line is an aid to the eye, illustrating the replication of a single origin. (**B**) Consistent with an adder model, the added length between birth and division is uncorrelated with length at birth (here and in all other scatter plots, the darker lines show the mean of the binned data and the error bars represent the standard error per bin). (**C**) The classical cell cycle is defined between consecutive division events, shown here with replication and division for slow growth conditions (i.e. without overlapping rounds of replication). (**D**) We introduce an alternative description framework where the cell cycle is defined between consecutive replication initiation events. The observables that are relevant to characterize the cell cycle in these two frameworks are indicated (see also *Table 1*).

The online version of this article includes the following source data and figure supplement(s) for figure 1:

**Source data 1.** Table with source data for *Figure 1B*.

**Figure supplement 1.** Schema of the cell cycle and variable definitions for the case of fast growth with overlapping replication cycles.

initiation timing, as origin splitting can be affected by a cohesion time (*Reyes-Lamothe et al., 2008*). These measurements allowed us to quantify each single cell cycle by a number of variables such as the growth rate, the sizes at birth, replication initiation, and division, the times between birth and replication initiation and the time between birth and division. As done previously, we assume that cell radius is constant and use cell length as a proxy for cell volume (*Adiciptaningrum et al., 2016*; *Taheri-Araghi et al., 2017*). Since we can follow cells over multiple generations, we can also measure quantities that span multiple division cycles such as the total time or total cell growth between consecutive replication initiation events. As we analyze growth conditions spanning cases with both single and overlapping rounds of replication, we defined two alternative ways of defining variables. In particular, while the cell cycle is classically defined from division to division (*Figure 1C*), we also use an alternative framework proposed recently (*Ho and Amir, 2015*; *Amir, 2017*), in which the cell cycle is defined from one replication initiation to the next (*Figure 1D*). As this framework is centered on origins of replication rather than on cells, we consequently define a new quantity $\Lambda$, the cell length per origin, which allows tracking of the amount of cell growth per origin of replication. For instance, in a case where a cell is born with an ongoing round of replication which started at time $t$, $\Lambda_i$ for that cell is defined as $\Lambda_i = L_i/4$ where $L_i$ is the length of the mother cell which contains four

origins at time $t$ (*Figure 1—figure supplement 1*). Using these definitions allows us to avoid having to apply artificial cut-offs as for example done in *Wallden et al. (2016)*. In this article, we explore a wide variety of models that take either a division-centric or replication-centric view of the cell cycle. Using specific correlations observed in our data, we show how entire classes of models can be rejected. In addition, we present a general statistical framework for ranking models based on their ability to capture the full correlation structure of the data.

## Cell size adder

We first verified whether our measurements support the previously observed adder behavior in cell size, and find that added length $dL$ between birth and division is indeed uncorrelated with length at birth $L_b$ in all growth conditions (*Figure 1B*), and, also in agreement with the adder model, the heritability of birth length between mother and daughter is characterized by a Pearson correlation coefficient of $r \approx 0.5$ (see *Appendix 1—table 1*). There has been a question in the literature as to whether adder behavior is also observed at slow growth. Slow growth conditions have only been tested in two studies. While an initial study reported that adder behavior is not observed at slow growth (*Wallden et al., 2016*), a more recent study did observe adder behavior at slow growth (*Si et al., 2019*). Our own observations are thus in line with this later study and it should also be noted that *Si et al. (2019)* only found a weak deviation from adder behavior when reanalyzing the data of *Wallden et al. (2016)*. Thus, while more independent studies are needed, on the whole the currently available data appear to support that adder behavior is independent of growth conditions.

## Replication initiation mass

A popular idea dating back to the 1960s and still often used today to explain the coupling of division and replication cycles is the initiation mass model. The observations that cell volume grows exponentially with growth rate (*Schaechter et al., 1958*) and that, across a range of conditions, the time between replication initiation and division is roughly constant (*Helmstetter et al., 1968*) led Donachie to propose that the volume per origin of replication is held constant (*Donachie, 1968*). In particular, the model proposes that initiation occurs when a cell reaches a critical volume. A simple prediction of this model is that, for a given cell, the cell length $L_i$ at which initiation occurs should be independent of other cell cycle variables such as the length at birth $L_b$. However, as can be seen in *Figure 2A*, we observe that the initiation length $L_i$ and birth length $L_b$ are clearly correlated in all conditions, rejecting the initiation mass model. The absence of an initiation mass has already been shown by *Adiciptaningrum et al. (2016)* and confirmed in a recent comprehensive analysis (*Micali et al., 2018a*). Note that, by assuming that the concentration of the molecule that controls

**Table 1.** Variables definitions.

| Division-centric | | Replication-centric | |
|---|---|---|---|
| ***Measured variables*** | | | |
| $L_b$ | Size at birth* | $\Lambda_i$ | Size per origin at *initial* replication initiation* |
| $L_d$ | Size at division* | $\Lambda_f$ | Size per origin at *final* replication initiation* |
| $T_{bd}$ | Duration between birth and division | $T_{if}$ | Duration between consecutive replication initiations |
| $L_i$ | Size at replication initiation* | $\Lambda_b$ | Size per origin at birth* |
| $T_{bi}$ | Duration between birth and replication initiation | $T_{ib}$ | Duration between replication initiation and birth |
| ***Derived variables*** | | | |
| $\lambda = \frac{1}{T_{bd}} \log L_d L_b$ | Cell growth rate* (between birth and division) | $\alpha = \frac{1}{T_{if}} \log \Lambda_f \Lambda_i$ | Cell growth rate* (between consecutive replication initiations) |
| $dL = L_d - L_b$ | Division 'adder' | $d\Lambda_{ib} = \Lambda_b - \Lambda_i$ | Replication 'adder' |
| $dL_{bi} = L_i - L_b$ | Birth-to-initiation 'adder' | $d\Lambda_{if} = \Lambda_f - \Lambda_i$ | Initiation-to-birth 'adder' |
| $R_{bd} = L_d / L_b$ | Growth ratio between birth and division | $R_{if} = \Lambda_f / \Lambda_i$ | Growth ratio between consecutive initiations |
| $R_{bi} = L_i / L_b$ | Growth ratio between birth and initiation | $R_{ib} = \Lambda_b / \Lambda_i$ | Growth ratio between initiation and birth |

* variables indicated by a star are measured from a linear fit of exponential elongation.

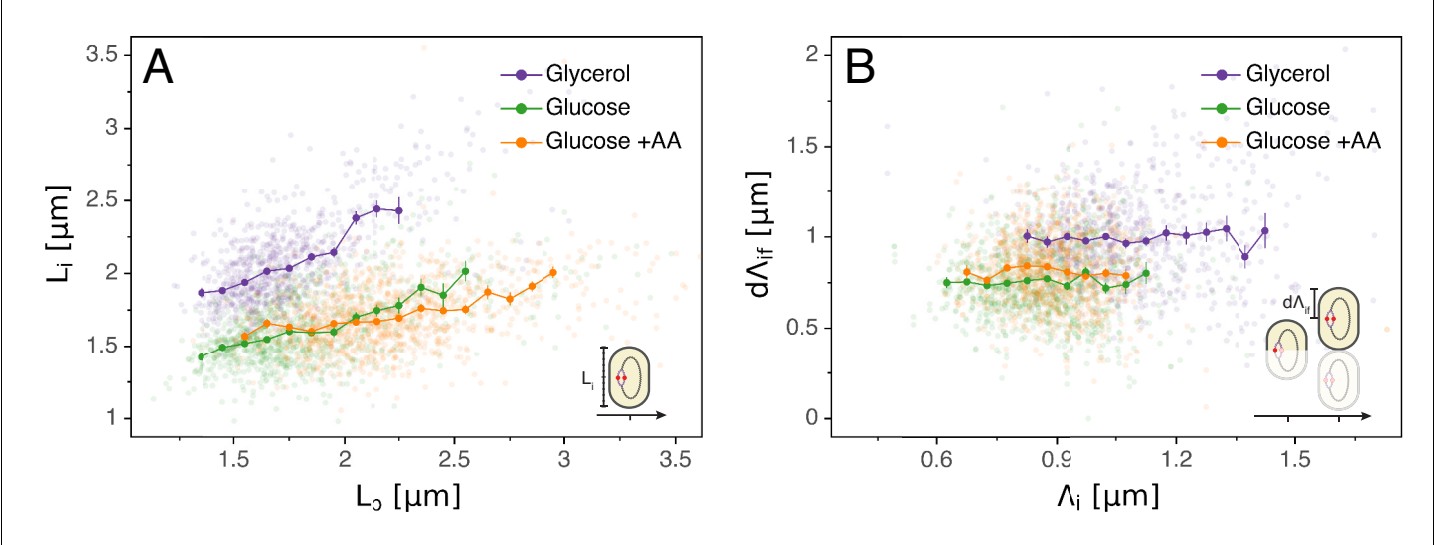

**Figure 2.** Models for initiation control. (A) The initiation mass model predicts that the length at initiation $L_i$ should be independent of the length at birth $L_b$. However, we observe clear positive correlations between $L_i$ and $L_b$ in all growth conditions. (B) In contrast, the length accumulated between two rounds of replication $d\Lambda_{if}$ is independent of the initiation size $\Lambda_i$, suggesting that replication initiation may be controlled by an adder mechanism. The online version of this article includes the following source data for figure 2:

**Source data 1.** Table with source data for *Figure 2*.

initiation is the same independent of growth rate, and a critical amount of this molecule has to accumulate per origin to trigger initiation, then this model would still predict that the average volume per origin in bulk is independent of growth rate (*Si et al., 2017*).

## Multiple origins accumulation model

Just as a constant average cell size can be accomplished by adding a constant volume per division cycle rather than by dividing at a critical division volume, so a constant average volume per origin of replication can also be implemented by controlling the added volume between replication initiations rather than by a critical initiation volume. A concrete proposal for such an adder mechanism, called the multiple origins accumulation model, has recently received increasing attention (*Ho and Amir, 2015*). In this model, a molecule that is expressed at a constant cellular concentration accumulates at each origin until it reaches a critical amount, triggering replication, after which it is degraded and starts a new accumulation cycle. Given that, for a molecule at constant concentration, the added volume over some time period is proportional to the amount produced of the molecule, the result of this process is that the cell adds a constant volume per origin $d\Lambda_{if}$ between initiation events (with $d\Lambda_{if} = \Lambda_f - \Lambda_i$ where indexes stand for 'initial' and 'final' respectively, see *Figure 1D* and *Table 1* for more details). Note also that in this 'per origin' formalism $d\Lambda_{if}$ is independent of the status of the division cycle, for example a cell might have two origins $n_{ori} = 2$ and a length at initiation $L_i^{ori2}$ or have already divided and have one origin $n_{ori} = 1$ and a length $L_i^{ori1}$, but $\Lambda_i = L_i^{ori2}/2 = (L_i^{ori2}/2)/1 = L_i^{ori1}/1$ is invariant. If replication is indeed triggered by such an adder mechanism, then one would expect the observed added lengths $d\Lambda_{if}$ to be independent of the length $\Lambda_i$ at the previous initiation. As shown in *Figure 2B*, our data support this prediction.

## Connecting replication and division cycles

Having validated the multiple origins accumulation model for replication control, we now investigate its relation to the division cycle. A common assumption is that the period $T_{id}$ from initiation to division (classically split into the replication period C and the end of replication to division period D) is constant and independent of growth rate (*Cooper and Helmstetter, 1968*; *Ho and Amir, 2015*). Such a constant period might suggest a timer mechanism by which, on average, a fixed time elapses in each cell between initiation and division. However, as visible in *Figure 3A*, while on average $T_{id}$ is

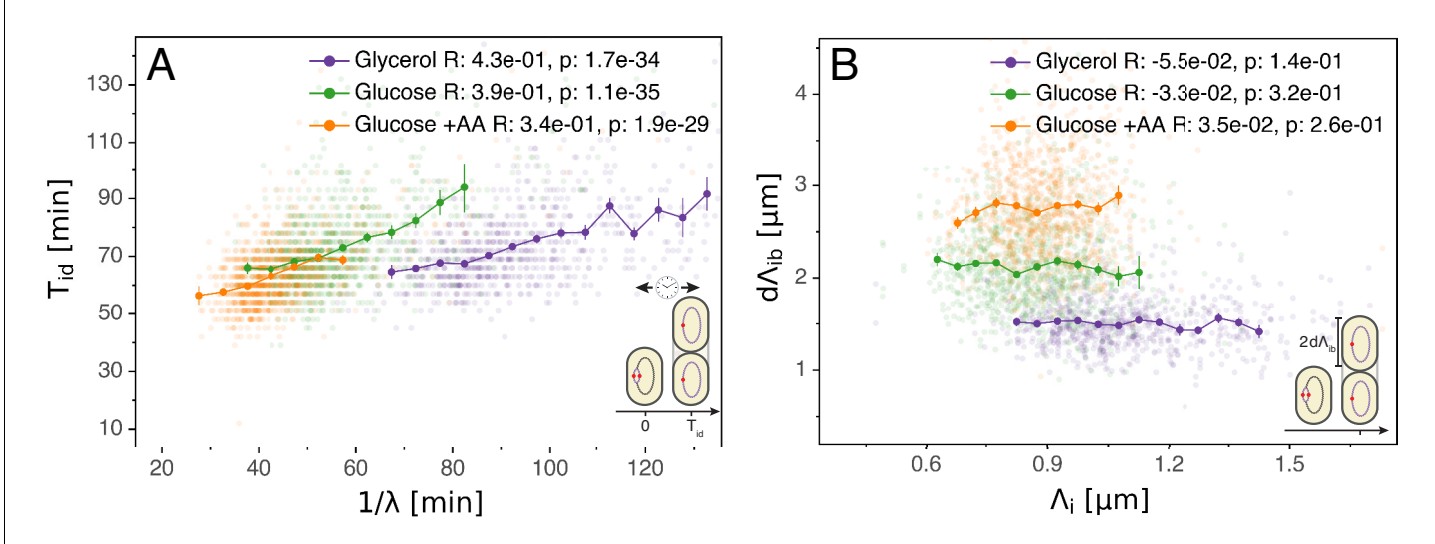

**Figure 3.** Initiation to division period. (**A**) Several models assume that a constant time passes from an initiation event to it corresponding division event. However, within each growth condition, that period is clearly dependent on fluctuations in growth rate. (**B**) The length accumulated from initiation to division is constant for each growth condition, suggesting an adder behavior for that period. In A and B, the Pearson correlation coefficient R and p values are indicated for each condition.

The online version of this article includes the following source data for figure 3:

**Source data 1.** Table with source data for *Figure 3*.

indeed rather constant across growth conditions, within each condition fast growing cells clearly complete this period faster than slow growing cells. One possibility is to assume that the time $T_{id}$ is coupled to growth rate $\lambda$ by an unspecified mechanism in such a way as to recover the empirically observed correlation (*Wallden et al., 2016*). However, *Figure 3B* reveals another and arguably simpler solution. We find that $d\Lambda_{id} = \Lambda_d - \Lambda_i$, the length per origin added by a cell between initiation and division, has an adder behavior as well: independently of its size at initiation $L_i$, a cell will complete the corresponding division cycle after having accumulated a constant volume per origin $d\Lambda_{id}$.

## The double-adder model

These observations motivated us to formulate a model in which the cell cycle does not run from one division to the next, but rather starts at initiation of replication, and in which both the next initiation of replication and the intervening division event, are controlled by two distinct adder mechanisms. In this replication-centric view, the cell cycles are controlled in a given condition by three variables: an average growth rate $\lambda$, an average added length per origin $d\Lambda_{if}$, and an average added length $d\Lambda_{id}$ between replication initiation and division. In particular, we assume that these three variables fluctuate independently around these averages for each individual cell cycle, and that all other parameters such as the sizes at birth, initiation, and the times between birth and division or between initiation and division, are all a function of these three fundamental variables. This double-adder model is sketched in *Figure 4* for the case of slow growth conditions: a cell growing at rate $\lambda$ initiates replication at length $L_i$ and thereby starts two adder processes. The cell then divides when reaching a size $L_d = L_i + n \, d\Lambda_{id} = n \, (\Lambda_i + d\Lambda_{id})$, where $n = 2$ is the number of replication origins. Second, the next replication round will be initiated when the total length has increased by $d\Lambda_{if}$ per origin.

## Simulations of the double-adder model

To assess to what extent our double-adder model can recover our quantitative observations, we resorted to numerical simulations. We first obtained from experimental data the empirical distributions of growth rates $\lambda$, the added length per initiation $d\Lambda_{if}$, and the added length between initiation and division $d\Lambda_{id}$. A series of cells are initialized at the initiation of replication, with sizes taken from the experimental distributions. For each cell, a growth rate $\lambda$ is independently drawn from its

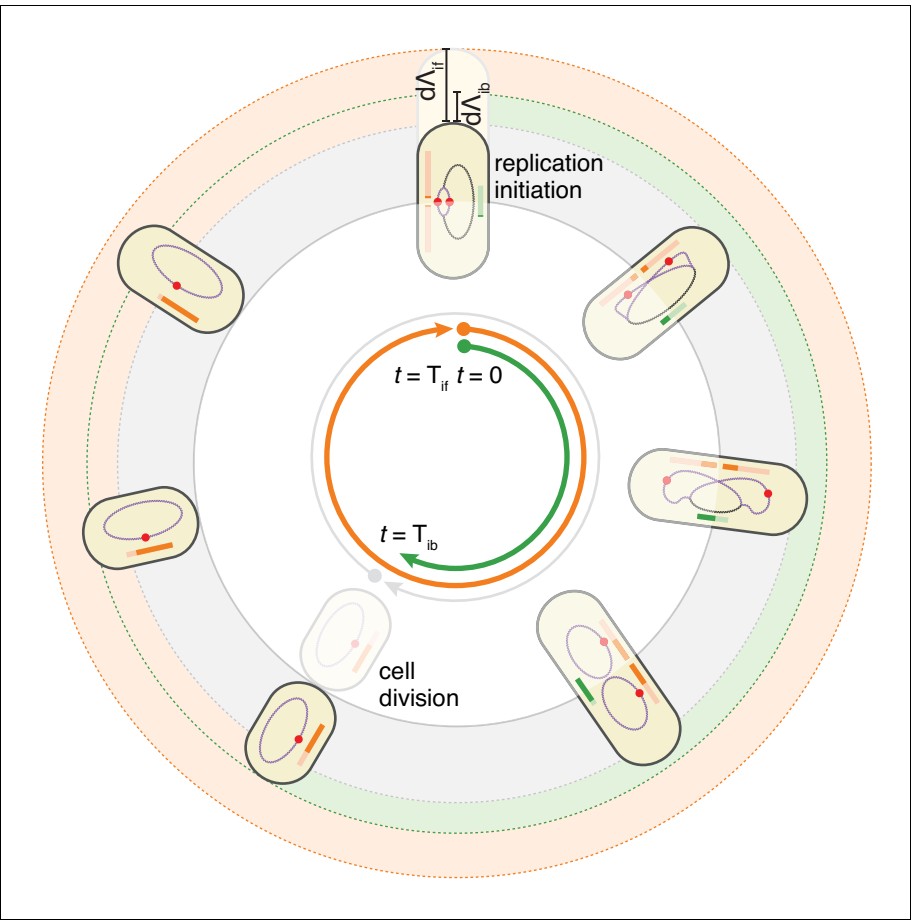

**Figure 4.** The double-adder model postulates that *E. coli* cell cycle is orchestrated by two independent adders, one for replication and one for division, reset at replication initiation. Both adders (shown as coloured bars) start one copy per origin at replication initiation and accumulate in parallel for some time. After the division adder (green) has reached its threshold, the cell divides, and the initiation adder (orange) splits between the daughters. It keeps accumulating until it reaches its own threshold and initiates a new round of division and replication adders. Note that the double-adder model is illustrated here for the simpler case of slow growth.

The online version of this article includes the following figure supplement(s) for figure 4:

**Figure supplement 1.** Average localization of the origin in cells growing in M9 glycerol.

empirical distribution, and values of $d\Lambda_{id}$ and $d\Lambda_{if}$ are drawn from independent distributions, to set the times of the next division and replication initiation events. This procedure is then iterated indefinitely, that is a new growth rate and values of each adder are independently drawn for each subsequent cycle. As has been observed previously (*Campos et al., 2014*) the growth rate is correlated ($r \approx 0.3$) between mother and daughter. Accounting for this mother-daughter correlation in growth rate was found not to be critical for capturing features of *E. coli* cell cycle, but was included in the model to reproduce simulation conditions of previous studies.

As can be seen in *Figure 5*, the double-adder model accurately reproduces measured distributions and correlations at all growth rates. In particular, the global adder behavior for cell size regulation naturally emerges from it (*Figure 5A*). Note however that in particular at fast growth (Glucose +AA), the relation between $dL$ and $L_b$ deviates from pure adder behavior, that is $dL$ weakly anti-correlates with $L_b$ (*Figure 1B*). Interestingly, this deviation is recovered as well in the simulations (*Figure 5A*), supporting the validity of our model. The specific relation between length at initiation $L_i$ and length at birth $L_b$, which prompted us to reject the initiation mass model, is too reproduced by the model (*Figure 5B*). Furthermore, the distribution of the number of origins at birth, which reflects the presence of overlapping replication cycles is also reproduced (*Figure 5D*). Finally, we

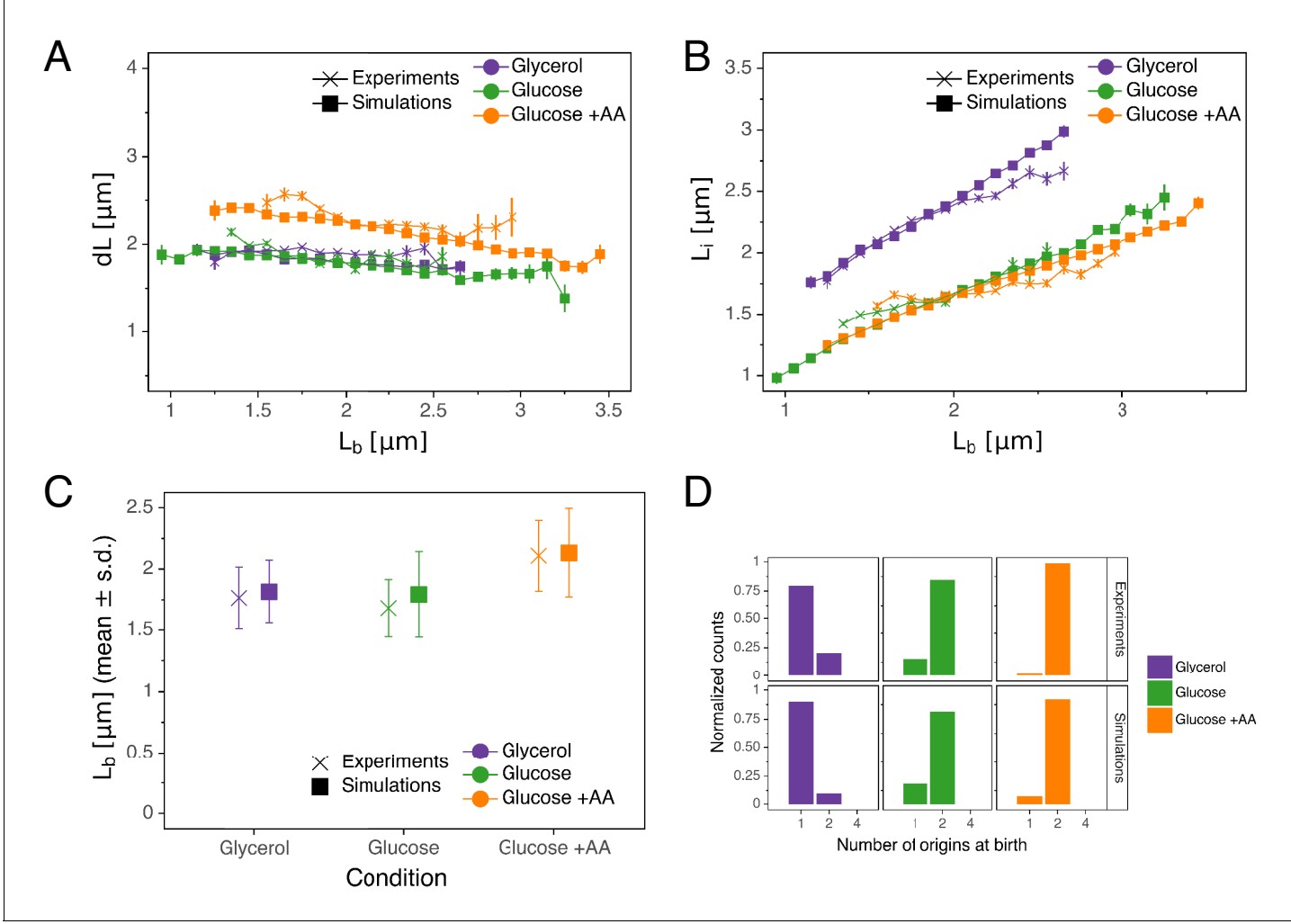

**Figure 5.** Comparison of predictions of the double-adder model with experimental observations. (**A**) Binned scatter plot of the added length between birth and division $dL$ versus length at birth $L_b$ shows no correlations in both the data and the simulations, demonstrating that the double-adder model reproduces the adder behavior at the level of cell size. (**B**) Binned scatter plot of the length at initiation $L_i$ versus length at birth $L_b$ shows almost identical correlations in data and simulation. (**C**) Average (± s.d) cell length at birth $L_b$. Both the mean and standard deviation are recovered in the model simulation. (**D**) The distribution of the number of origins at birth is also highly similar between experiments and data for all growth conditions. The online version of this article includes the following source data and figure supplement(s) for figure 5:

**Source data 1.** Table with source data for *Figure 5AB*.
**Source data 2.** Table with source data for *Figure 5C*.
**Source data 3.** Table with source data for *Figure 5D*.
**Figure supplement 1.** Detailed comparisons of cell cycle variables distributions and correlations between experiments and simulations for M9+glycerol condition (with automated origin tracking).
**Figure supplement 2.** Detailed comparisons of cell cycle variables distributions and correlations between experiments and simulations for M9+glycerol condition (with manual origin tracking).
**Figure supplement 3.** Detailed comparisons of cell cycle variables distributions and correlations between experiments and simulations for M9+glucose condition (with manual origin tracking).
**Figure supplement 4.** Detailed comparisons of cell cycle variables distributions and correlations between experiments and simulations for M9+glucose +8a.a. condition (with manual origin tracking).

note that while the average cell length parameters (e.g. for $L_b$ in *Figure 5C*) are recovered in the simulations, the variances of the simulated distributions are a bit larger in the faster growth conditions, which can be attributed to an overestimation of the variances of the two adder distributions. An exhaustive comparison between experiments and simulations can be found in *Figure 5—figure supplement 1* through *Figure 5—figure supplement 4*.

As our model recovers the classic adder from birth to division, that is independence of $L_b$ and $dL$, one could ask whether, instead of assuming division to be controlled by an adder between initiation and birth, one could simply have a classic birth-to-division adder running independently and in parallel with a replication adder. *Harris and Theriot (2016)* proposed that a homeostatic mechanism maintaining a constant cell surface to volume ratio could produce a birth-to-division adder, but only if division were controlled completely independently of replication. A recent study that emerged during the course of this work (*Si et al., 2019*) also argued that the division and replication cycles are uncoupled and controlled independently. To test these ideas, we implemented a model in which division and replication are controlled by completely independent adder mechanisms. Although this model recapitulates most experimental observations, it predicts a negative correlation between volume at initiation $L_i$ and the added volume $d\Lambda_{ib}$ between initiation and birth (Appendix 2), which is at odds with the adder behavior we observe in the data (*Figure 3*). It should also be noted that purely geometric division models such as the *Harris and Theriot (2016)* model have been recently challenged more broadly by research indicating that surface expansion is coupled to dry-mass increase rather than to volume increase (*Oldewurtel et al., 2019*).

Recently, *Micali et al. (2018b)* introduced a wider class of models in which the replication and division cycles are controlled by two concurrently running processes, which are coupled through a check-point that demands a certain minimal amount of time has to elapse between replication initiation and division. This class includes models in which independent replication and division adders are coupled through such a check-point and, if parameters are appropriately tuned, it is possible for such a model to exhibit no correlation between $L_i$ and $d\Lambda_{ib}$, that is reproduce the observed adder behavior for the period between replication initiation and division (Micali and Cosentino Lagomarsino, personal communication). Although such a model is thus potentially consistent with our observations, it requires the parameters to be precisely tuned and, moreover, this tuning has to apply to all growth conditions.

## The double-adder model best captures the correlation structure of the data

In this work, we used empirically observed correlations between certain variables, such as those shown in *Figure 2* and *Figure 3*, to motivate our double-adder model, and then used numerical simulations of this model to show that this model also successfully reproduces other features of the experimental data (*Figure 5*). However, it is conceivable that other models would be equally successful in reproducing the experimental observations. We thus aimed to devise a general approach for systematically comparing the performance of a large class of models.

The basic idea behind this approach is that, if the cell cycle is controlled by independent control processes, then we expect the variables associated with these control processes to fluctuate independently. To illustrate this, let's consider for the moment the simpler case of the division cycle and cell size control (*Figure 6A*). If cell size homeostasis were controlled by a 'sizer' mechanism, we would expect the size at division $L_d$ to fluctuate independently of size at birth $L_b$ and growth rate $\lambda$. If it were instead controlled by a 'timer', then we would expect the time $T_{bd}$ between birth and division to fluctuate independently from size at birth $L_b$ and growth rate $\lambda$. However, the measurements show that the time $T_{bd}$ is clearly negatively correlated with both growth rate and size at birth. Instead, the added volume $dL$ fluctuates almost independently of size at birth $L_b$ and growth rate $\lambda$ (*Figure 6A*). It is precisely this independence of fluctuations in added volume from fluctuations in the other variables, that constitutes the support of the adder mechanism. Here, we generalize this idea by systematically considering all sets of variables that can be used to describe the cell cycle in single cells, and quantifying, for each set of variables, the extent to which they fluctuate independently.

A single division cycle needs 3 variables to be fully described and we will refer to such sets of variables as decompositions. For example, the set $(L_b, L_d, \lambda)$ corresponds to a 'sizer' decomposition, the set $(L_b, T_{bd}, \lambda)$ to a 'timer' decomposition, and $(L_b, dL, \lambda)$ to an adder decomposition. For each decomposition, we can now calculate, from the data, the matrix $R$ of observed correlations $R_{ij}$ between each pair of variables $(i, j)$ in the decomposition. For example, *Figure 6A* shows the correlation matrices $R$ for the adder and timer decompositions. The more independent the variables are in the decomposition, the smaller the correlations will be. As a measure for the overall independence of the variables in a decomposition we use the determinant of the matrix $R$, that is we define

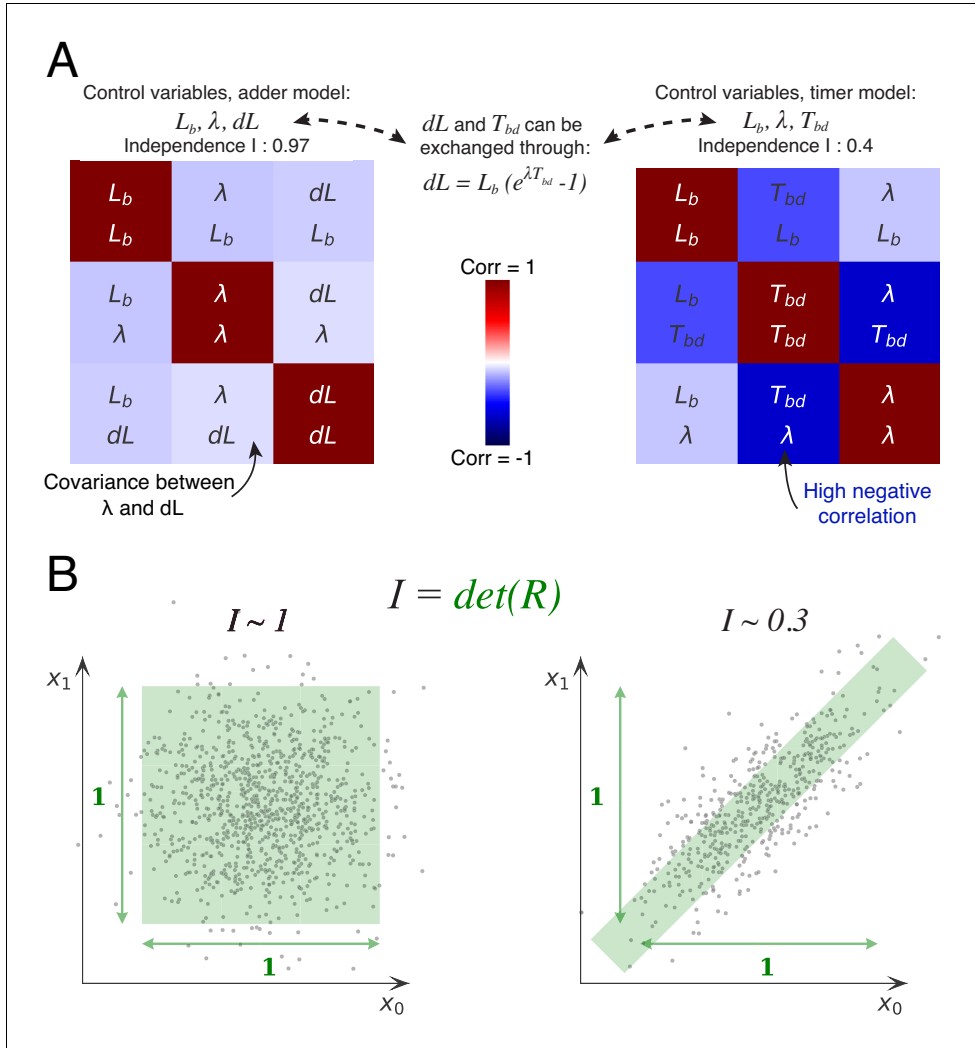

**Figure 6.** Decomposition method. (**A**) The cell division cycle of a single cell can be described by different combinations of three variables which we refer to as decompositions. For example, the set $(L_b, dL, \lambda)$ corresponds to an adder decomposition, while $(L_b, T_{bd}, \lambda)$ corresponds to a timer decomposition. For each possible decomposition, we can calculate the matrix $R$ of observed correlations $R_{ij}$ between each pair of variables in the decomposition from the data. Shown are the correlation matrices for the adder decomposition (left) and the timer decomposition (right), with positive correlations shown in red and negative correlations in blue. The independence $I$ of each decomposition is defined as the determined of the correlation matrix $I = \det(R)$ and is indicated on top of each matrix. While the independence for the adder decomposition is close to the possible maximum of 1, the independence of the timer correlation is much lower due to a strong negative correlation between growth rate $\lambda$ and the cell cycle duration $T_{bd}$. (**B**) Conceptual illustration of the independence measure $I$. For each decomposition, the data can be thought of as a scatter of points in the space of the decomposition's variables, normalized such that the variance of points along each dimension is 1. In this conceptual example, we show two scatters of points for the two variables $x_0$ and $x_1$. The independence $I$ corresponds to the square of the volume covered by the scatter of points. On the left, there is virtually no correlation between the two variables, that is $R_{01} \approx 0$, such that the independence $I = \det(R) = 1 - R_{01}^2 \approx 1$. In contrast, on the right there is a strong correlation, leading to a much lower independence $I \approx 0.3$. In this way, the independence measure quantifies to what extent the variables in the decomposition fluctuate independently, and this measure applies to scatters of any number of dimensions.

The online version of this article includes the following source data and figure supplement(s) for figure 6:

**Source data 1.** Table with source data for *Figure 6*.

**Figure supplement 1.** Correlation matrices for all decompositions of the division cycle.

$I = \det(R)$. The intuitive meaning of this measure is illustrated in *Figure 6B*. We can think of each cell cycle as a point in the three-dimensional space spanned by the decomposition, such that all cell cycle observations form a scatter of points in this space. If we normalize each axis by setting its scale such that the variance equals 1 along the axis, then the independence $I = \det(R)$ corresponds exactly to the *volume* covered by the scatter of points. Note that, if there are no correlations, the matrix $R$ will become the identity matrix, that is $R_{ii} = 1$ and $R_{ij} = 0$ when $i \neq j$, and the independence will become exactly 1. The larger the correlations between the variables, the smaller the volume $I$ will become, and it will become zero in the limit of perfect correlation between 2 or more variables. Thus, the independence per definition ranges from 0 to 1 (perfect independence).

We can now apply this systematic approach to the complete cell cycle case. In this case, we measured the variables shown in *Table 1* (*measured variables*) either for the replication- or the division-centric view of the cell cycle. As for the division cycle case analyzed in *Figure 6*, we can use equations relating these variables in order to derive a set of additional variables as shown in *Table 1* (*derived variables*). We need four variables to define both replication and division cycles, and using now all the measured and derived variables, as well as the equations connecting them, we can create all possible decompositions of four variables sufficient to describe the cell cycle, and measure their independence using the decomposition method. Finally, we can rank all decompositions according to their independence to find which ones offer the most accurate description of the data. Such a statistical analysis is only relevant when applied to a large dataset and we therefore focus here on the slow growth condition (M9 glycerol) for which we implemented automatic origin tracking.

The tables in *Figure 7A* show the five best models ranked by decreasing independence (all decompositions can be found in *Figure 7—figure supplement 3*). Note that these variable sets include previously proposed sizer and timer models as special cases, for example the inter-initiation model combined with an initiation to division timer (*Ho and Amir, 2015*) is highlighted in red in *Figure 7—figure supplement 3*. The most successful decompositions are shown in greater detail as correlation matrices *Figure 7B*. We find in general that none of the division-centric models accomplishes high independence. For example, as shown in the covariance matrix of *Figure 7B* right, the best division-centric model is plagued by high correlation between $L_b$ and $dL_{bi}$. This strongly suggests that the cell cycle control is better described from a replication-centric point of view. Of all replication-centric models, our double-adder model clearly reaches the highest independence, followed by various derivative models in which one of the adders is replaced by another variable. Notably, the top three decompositions are the same for the real data and for the data from the simulations of the double-adder model, underscoring that these near optimal variants are expected for data produced by a double-adder mechanism. We note that independence of our double-adder model on the real data is a bit lower than on simulated data *Figure 7B,* that is 0.88 versus 0.98. This residual dependence might either result from correlated errors in the measurements, or it might reflect some small biological dependence not captured by our model. As an additional control, we also applied our decomposition analysis to a dataset from a simulation with the multiple origins accumulation model in which there is an inter-initiation added volume $d\Lambda_{if}$ and a timer from initiation to division $T_{ib}$. As shown in *Figure 7—figure supplement 4*, the decomposition analysis successfully identifies the model used in the simulation as having the highest Independence $I$. Finally, to verify that the results obtained by analyzing the fluctuations of the division and replication cycles are not primarily driven by differences in experimental details (such as strains or reporters used to measure replication initiation), we also applied our decomposition method to a dataset from a study that appeared in the course of this work (*Si et al., 2019*). As shown in *Figure 7—figure supplement 5*, here again the double-adder model is the most successful at explaining the experimental measurements. In summary, this systematic analysis shows that, within a large class of alternative models, the double-adder model best captures the correlation structure of both our data and data recently obtained by another laboratory.

## Discussion

Thanks to experimental techniques like the one used here, models of bacterial cell cycle regulation dating back to the 1960s have been recently re-examined in detail in several studies (*Campos et al., 2014*; *Tanouchi et al., 2015*; *Ho and Amir, 2015*; *Adiciptaningrum et al., 2016*; *Wallden et al., 2016*; *Si et al., 2017*; *Logsdon et al., 2017*; *Micali et al., 2018a*; *Eun et al., 2018*; *Si et al., 2019*).

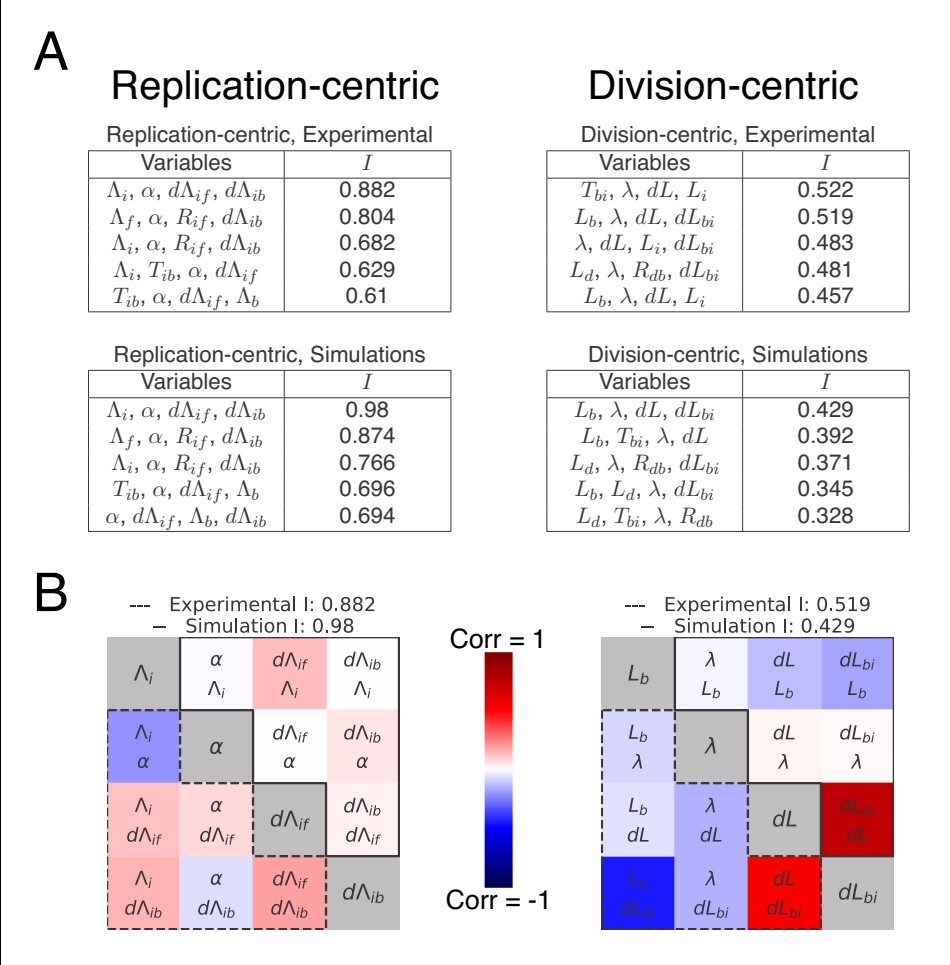

**Figure 7.** Decomposition analysis applied to the division and replication cycles. (**A**) The tables show the independence measures $I$ for the top scoring decompositions of the division and replication cycles. In these tables, each line represents a possible decomposition and its independence $I$. As there are two ways to see the cell cycle (replication- and division-centric), we present decompositions for both replication- (left) and division-centric decompositions (right). In addition, we show the top decompositions both for the correlation matrices of the experimental data of the growth conditions M9+glycerol (automated analysis) and for the data from the simulations of the double-adder model (top and bottom rows, respectively). Results for the full list of decompositions can be found in *Figure 7—figure supplement 3*. Note that the decomposition analysis clearly identifies the replication-centric double-adder characterized by $\Lambda_i$, $\alpha$, $d\Lambda_{if}$ and $d\Lambda_{ib}$ as the best decomposition. The fact that the double-adder decomposition is also top scoring (with $I \simeq 1$) for data from the simulation of the double-adder confirms that the decomposition analysis works as expected. (**B**) Correlation matrices for the best decompositions for replication-centric (left) and division-centric models (right). As in *Figure 6A*, each matrix represents one decomposition, and each element of the matrix shows the correlation of the two variables indicated within it. The level and sign of correlation is given by the color bar. As the lower left and upper right triangles of the matrices are redundant, we use them to show correlations from both experimental and simulation data in a single matrix. The lower-left corners bounded by a dotted line contain correlations from experimental data and the upper-right ones, bounded by a continuous line, from simulation data. The diagonal summarizes the set of variables. The best replication-centric model (left) has only weak correlations between its variables as reflected in high independence, while the best division-centric model has a few highly correlated variables leading to low independence.

The online version of this article includes the following source data and figure supplement(s) for figure 7:

**Source data 1.** Table with source data for replication-centric decompositions of both experimental and simulation data of *Figure 7* and *Figure 7—figure supplement 1*, *Figure 7—figure supplement 2* and *Figure 7—figure supplement 3*.

*Figure 7 continued on next page*

*Figure 7 continued*

**Source data 2.** Table with source data for division-centric decompositions of both experimental and simulation data of *Figure 7* and *Figure 7—figure supplement 1*, *Figure 7—figure supplement 2* and *Figure 7—figure supplement 3*.

**Figure supplement 1.** Correlation matrices for the top nine decompositions of the experimental data.

**Figure supplement 2.** Correlation matrices for the top nine decompositions of the data from simulations of the double-adder model.

**Figure supplement 3.** Full list of independences *I* for all replication-centric and division-centric models.

**Figure supplement 4.** Top scoring decompositions for data from simulations of an alternative model.

**Figure supplement 4—source data 1.** Table with source data for replication-centric decompositions of *Figure 7—figure supplement 4*.

**Figure supplement 4—source data 2.** Table with source data for division-centric decompositions of *Figure 7—figure supplement 4*.

**Figure supplement 5.** Top scoring decompositions for data from *Si et al. (2019)*.

**Figure supplement 5—source data 1.** Table with source data for replication-centric decompositions of *Figure 7—figure supplement 5*.

**Figure supplement 5—source data 2.** Table with source data for division-centric decompositions of *Figure 7—figure supplement 5*.

Although these data have shed important new light on the regulation of bacterial physiology, such as identifying the adder behavior in cell growth and division, it has remained somewhat unclear to what extent different models are consistent with the data. In particular, many studies only focus on certain correlations and dependencies between measurable cell cycle quantities, and these can often be explained by multiple models. In this study, we have in a first step empirically built a model which is based on previous ideas and which recapitulates measured cell cycle parameters. This model makes replication the central regulator of the cell cycle, with each initiation round triggering subsequent division and replication events through concurrent adder processes. In a second step, we then designed and applied a statistical method to determine, within a class of models built on biologically relevant cell cycle variables, which set of those variables best explains the correlation structure observed in the measurements. Following this fully independent and more systematic approach, our empirical double-adder model clearly comes out as the most successful. It should be noted that, whereas models based on a timer between initiation and division naturally predict the well-known exponential increase of average cell-size with average growth rate measured in the bulk (*Schaechter et al., 1958*; *Taheri-Araghi et al., 2017*), the double-adder does not require such a relationship. However, neither is such a relationship inconsistent with the double-adder model. The difference between a timer and adder model is in the nature of the correlations of different cell cycle quantities across single cells within one condition, and not in the behavior of their averages across conditions. Therefore, it is possible to implement the same dependence of average size on growth rate using different models that have different single-cell correlation structure. In particular, in order for the average bulk cell-size cell size to grow exponentially with the average bulk growth rate, it is enough that the *average* time between initiation and division is independent of growth rate and this can also be achieved with an adder model. For example, if we imagine that the initiation to division adder is implemented by the accumulation of a key molecule to a critical amount, and we assume that the average rate of accumulation of this molecule is independent of the bulk growth rate, then the average cell-size will grow exponentially with growth rate. At the same time, within such an adder model the single-cell fluctuations in the added volume will be independent of the single-cell fluctuations in growth rate and initiation size, as required by our observations, and in contrast to the predictions of a timer model, which would predict fluctuations in growth rate and added volume to correlate positively.

While the division and replication cycles are seemingly coupled, our analysis demonstrates that two simple adders connected to replication initiation are sufficient to recapitulate both cycles without explicitly enforcing constraints reflecting mechanisms such as over-initiation control by SeqA and nucleoid occlusion which ensures that division only occurs after chromosome replication is completed. The initiation-to-initiation adder mimics SeqA activity by creating a refractory period without initiation, and the initiation-to-division adder ensures that a minimal time is allocated for replication

to complete. While the simulations might in rare cases generate unrealistic situations, for example if a large initiation adder is combined with a small division adder leading to premature division, those clashes seem rare enough to not affect the global statistical behavior of the model. Naturally, the model would break down and these controls would need to be explicitly included in the case where cells are subject to stress conditions where these control mechanisms act as fail-safes for example to ensure that division is delayed if DNA repair is needed. In the course of our study, new research proposed that division and replication cycles are only seemingly connected, and used perturbation methods to drive cells to states where the uncoupling is revealed (*Si et al., 2019*). While such perturbation studies are very informative, more work is needed to understand to what extent hidden compensatory mechanisms might be at play when affecting DnaA or FtsZ expression. Also, that study focuses exclusively on a model which explicitly enforces various correlations between variables unlike our model which naturally produces such relations. Another recent study (*Wehrens et al., 2018*) using a perturbative approach has shown that cells driven to a filamentous state recover normal sizes through successive single division events seemingly independently from initiation and following an adder behaviour. As division inhibition also hinders replication related processes such as chromosome segregation (*Sánchez-Gorostiaga et al., 2016*), it is difficult to untangle the different mechanisms that might be at play. Complementing such experiments with a monitoring of replication as done here would surely be a worthwhile future endeavour, a task though which is beyond the purpose of this study, which tries to clarify the normal growth case.

Interestingly, a double-adder mechanism similar to the one that we propose here has been shown to explain cell cycle control in mycobacteria (*Logsdon et al., 2017*). These mycobacteria have a much more complex cell cycle than *E. coli*, in particular characterized by a strong asymmetry between daughter cells and a growth rate almost an order of magnitude smaller than that of *E. coli*. Despite those important differences, it was shown that mycobacterial cell cycles exhibit adder behavior for both division and replication starting at initiation, in a manner highly similar to our observations in *E. coli*. This suggests that the mechanism connecting replication and division must be quite fundamental and independent of the specifics of available genes and their expression.

Although the single-cell observations provide clear indications of which variables are most likely to be directly involved in the cell cycle control, they of course do not indicate the underlying molecular mechanisms. However, it is not hard to speculate about possible molecular mechanisms that could implement the double-adder behavior. As others have pointed out previously (*Ho and Amir, 2015*), an adder for the regulation of replication initiation can be easily implemented at the molecular level by having a 'sensor' protein that builds up at each origin, and that triggers replication initiation whenever a critical mass is reached at a given origin. If this sensor protein is additionally homeostatically controlled such that its production relative to the overall protein production is kept constant, then the average volume per origin will also be kept constant across conditions.

It is more challenging to define a molecular system that can implement the second adder that controls division. The main challenge is that this adder does not run throughout the entire cell cycle, but only between replication initiation and division. It is well known that division is driven by the polymerization of the FtsZ ring, which includes a host of other FtsZ-ring associated proteins, and its progressive constriction. It might seem simplest to assume that the division adder could be implemented directly through FtsZ production, again in the logic of the regulated 'sensor' mentioned above. However, this would require FtsZ to be produced and accumulating at the division sites only from replication initiation to cell division. Although this is conceivable, that is it is known that FtsZ and other division proteins are heavily regulated at several levels (*Dewar and Dorazi, 2000*) and that especially in slow growth conditions its concentration varies during the cell cycle (*Männik et al., 2018*), it is hard to imagine how this model could work under fast growth conditions in which there are overlapping rounds of replication such that FtsZ would be constantly expressed. Moreover recent data (*Si et al., 2019*) rather suggest that FtsZ concentration is constant during the cell cycle.

Alternatively, rather than FtsZ production, Ftsz polymerization could be regulated. One remarkable observation that is well known within the field (*Lau et al., 2004*; *Nielsen et al., 2006*) and that we also observe in our data (see *Figure 4—figure supplement 1*), is that origins always occupy the position of future division sites (mid-cell, 1/4 and 3/4 positions etc.) when replication is initiated. This observation not only suggests that, at replication initiation, some local molecular event occurs that will eventually trigger division at the same site, but it is also remarkably consistent with the idea of an adder running only between replication initiation and division. One long-standing idea that is

consistent with these observations is that some molecular event that occurs during replication initiation triggers the start of FtsZ ring formation, and that the timing from initiation to division is controlled by the polymerization dynamics of the FtsZ ring (*Weart and Levin, 2003*). At the molecular level, the common triggering of initiation and polymerization might be explained by the accumulation of acidic phosholipids in the cell membrane precisely at future division sites (*Renner and Weibel, 2011*) where they probably interact with components of the division machinery. At the same time those lipids are known to play a role in promoting replication by rejuvenating the initiator protein DnaA-ADP into DnaA-ATP (*Saxena et al., 2013*), and might therefore be a 'hub' coordinating the two cycles. Finally, it remains to be explained how FtsZ polymerization or pole building could result in an adder behavior. For that purpose, future experiments should focus on combining the type of information collected in this study and detailed measures of the dynamics of FtsZ-ring assembly and constriction as done in *Coltharp et al. (2016)*.

## Materials and methods

### Bacterial strains and media

All strains are derived from the K-12 strain BW27378, a Δ(araH-araF)570(::FRT) derivative of the Keio collection background strain (*Baba et al., 2006*) obtained from the Yale Coli Genetic Stock Center. This strain was further modified by λ-Red recombination (*Datsenko and Wanner, 2000*) and P1 transduction to result in ΔaraFGH(::FRT), ΔaraE(::FRT), ΔlacIZYA(::FRT). A 250 lacO repeats FROS array with chloramphenicol resistance was inserted close to the origin of replication in the *asnA* gene by λ-Red recombination and P1 transduction resulting in strain GW273. The lacO-CmR array was derived from the original plasmid pLau43 (*Lau et al., 2004*) by replacing the kanamycin resistance and a series of operators on both sides of it with the CmR gene. For visualization of the array, GW273 was transformed with plasmid pGW266 expressing a LacI-mVenus fusion, resulting in strain GW296. The plasmid is derived from the original FROS plasmid pLAU53 (*Lau et al., 2004*) from which the tetR construct was removed and the lacI-CFP replaced with lacI-mVenus. For the experiment analyzed automatically, strain GW296 was additionally transformed with plasmid pGW339 expressing FtsZ-mVenus under the control of the araBAD promoter using 0.002% arabinose for induction, resulting in strain GW339. Expression is tightly controlled by using the approach proposed in *Morgan-Kiss et al. (2002)*.

All experiments were done using M9 minimal media supplemented with 2mM MgSO4, 0.1mM CaCl2, and sugars (0.2% for glucose and 0.2% for glycerol). In one experiment, the media was supplemented with eight amino acids at a concentration of 5 μg/mL$^{-1}$: Threonine, Aspagrinine, Methionine, Proline, Leucine, Tryptophane, Serine, Alanine. All experiments were carried out at 37°C.

### Microfluidic device fabrication

Mother Machine experiments were performed using the Dual Input Mother Machine (DIMM) microfluidic design which has been described elsewhere (*Kaiser et al., 2018*) and is freely available online (https://metafluidics.org/devices/dual-input-mother-machine/); since no change of conditions was intended during experiments, the same media was flown at both inputs.

Several microfluidics masters were produced using soft lithography techniques by micro-resist Gmbh; two masters with regular growth channels of suitable size ( 0.8 μm width × 0.9 μm height for growth in glycerol, and 1 μm width × 1.2 μm height for growth in glucose) were used for all experiments.

For each experiment, a new chip was produced by pouring PDMS (Sylgard 184 with 1:9 w/w ratio of curing agent) on the master and baking it for 4 hrs or more at 80°C. After cutting the chip and punching inlets, the chip was bonded to a #1.5 glass coverslip as follows: the coverslip was manually washed in water and soap, rinsed in isopropanol then water; the chip cleaned from dust using MagicTape, rinsed in isopropanol then water; surfaces were activated with air plasma (40 s at 1500 μm of Hg) before being put in contact; the assembled chip was cooked 1 hr or more at 80°C.

Before running the experiment, the chip was primed and incubated 1 hr at 37°C using passivation buffer (2.5 mg/mL salmon sperm DNA, 7.5 mg/mL bovine serum albumin) for the mother machine part and water for the overflow channels.

## Experiment setup and conditions

Bacteria were stored as frozen glycerol stocks at −80°C and streaked onto LB agar plates to obtain clonal colonies. Overnight precultures were grown from single colonies in the same growth media as the experiment. The next day, cells were diluted 100-fold into fresh medium and harvested after 4–6 hr.

The experimental apparatus was initialized, pre-warmed and equilibrated. Media flow was controlled using a pressure controller and monitored with flow-meters, set to run a total flow of $\approx$ 1.5 L/min (corresponding to a pressure of $\approx$ 1600 mbar).

The primed microfluidic chip was mounted, connected to media supply and flushed with running media for 30 min or more to rinse passivation buffer. The grown cell culture was centrifuged at 4000× g for 5 min, and the pellet re-suspended in a few μL supernatant and injected into the device from the outlet using the pressure controller. To facilitate the filling of growth channels by swimming and diffusing cells, the pressure was adjusted in order to maintain minimal flow in the main channel (loading time $\approx$ 40 min).

After loading, bacteria were incubated during 2 hr before starting image acquisition. Every 3 min, phase contrast and fluorescence images were acquired for several well-separated positions in parallel.

## Microscopy and image analysis

An inverted Nikon Ti-E microscope, equipped with a motorized xy-stage and enclosed in a temperature incubator (TheCube, Life Imaging Systems), was used to perform all experiments. The sample was fixed on the stage using metal clamps and focus was maintained using hardware autofocus (Perfect Focus System, Nikon). Images were recorded using a CFI Plan Apochromat Lambda DM ×100 objective (NA 1.45, WD 0.13 mm) and a CMOS camera (Hamamatsu Orca-Flash 4.0). The setup was controlled using microManager (*Edelstein et al., 2014*) and time-lapse movies were recorded with its Multi-Dimensional Acquisition engine. Phase contrast images were acquired using 200 ms exposure (CoolLED pE-100, full power). Images of mCherry fluorescence were acquired using 200 ms exposure (Lumencor SpectraX, Green LED at 33% with ND4) using a Semrock triple-band emission filter (FF01-475/543/702-25).

Image analysis was performed using MoMA (*Kaiser et al., 2018*) as described in its documentation (https://github.com/fjug/MoMA/wiki). Raw image datasets were transferred to a centralised storage and preprocessed in batch. Growth channels were chosen randomly after discarding those where cell cycle arrest occurred in the mother cell, and curated manually in MoMA. An exponential elongation model was then fitted to each cell cycle, and cycles presenting large deviations were discarded (1–3% of each experiment, see Appendix 1).

For the automated origin detection and tracking, we used custom Python code (*Witz, 2019*; copy archived at https://github.com/elifesciences-publications/DoubleAdderArticle) which makes extensive use of the packages numpy (*van der Walt et al., 2011*), scipy (*Jones et al., 2001*), matplotlib (*Hunter, 2007*), pandas (*McKinney, 2010*), and scikit-image (*van der Walt et al., 2014*). Spots were detected following the method proposed in *Aguet et al. (2013)*. Briefly, amplitude and background are estimated for each pixel using a fast filtering method and a spot model corresponding to the optical setup. Among the local maxima found in the amplitude estimates, spots are then selected using a statistical test based on the assumption that background noise is Gaussian. To track spots, we used the trackpy package (*Allan et al., 2018*). Cell cycles with incoherent properties (e.g. missing origin because of a failed detection) were discarded (~14%). The time of initiation was assigned as the first time point where a track splits into two. For the manual analysis of the other experiments, the frame showing origin splitting was selected manually.

Using the timing of origin splitting, the corresponding cell length could be determined. All further variables like $dL$ or $d\Lambda_{ib}$ are deduced from the primary variables. For the decomposition analysis, a pseudo-cell cycle was created by concatenating the mother cell cycle from initiation to division with the daughter cell cycle from birth to initiation (the operation is repeated for both daughters separately). The growth rate $\alpha$ for this pseudo-cell cycle was again obtained by fitting an exponential growth model. All the growth lanes corresponding to a given conditions were then pooled to generate the various statistics shown in this article.

The entire analysis pipeline is available as Python modules and Jupyter Notebooks on Github, https://github.com/guiwitz/DoubleAdderArticle (*Witz, 2019*).

## Simulations

The numerical implementation of the model described in *Figure 4* and used in *Figure 5* requires several parameters for each individual cell cycle. To generate those, the following distributions were extracted from experimental data, and if needed their means and variances were obtained by a fitting procedure:

- The growth rate distributions $P(\lambda)$.
- The growth rate correlation from mother to daughters.
- The length distributions of the two adder processes $P(d\Lambda_{ib})$ and $P(d\Lambda_{if})$.
- The distributions of length ratios between sister cells to account for imprecision in division placement $P(r)$.

For the simulation, a series of 500 cells is initialized with all required parameters: initial length $L_0$ taken from the birth length distribution, $\lambda = P(\lambda)$, number of origins $n_{ori} = 1$, and the two adders $d\Lambda_{ib} = P(\lambda)$ and $d\Lambda_{if} = P(d\Lambda_{if})$ whose counters are starting at 0. The exact initialization is not crucial as the system relaxes to its equilibrium state after a few generations. Cells are then growing incrementally following an exponential law, and the added length is monitored. Every time the cell reaches its target $d\Lambda_{if}$, the number of origins doubles and a new initiation adder is drawn from $P(d\Lambda_{if})$. Every time the cell reaches its target $d\Lambda_{ib}$ the cell (1) divides into two cells using a division ratio drawn $P(r)$, (2) the number of origins per cell is divided by two, (3) a new division adder is drawn from $P(d\Lambda_{ib})$, and finally (4) a new growth rate is drawn from $P(\lambda)$. Each simulation runs for 30 hrs in steps of 1 min. In the end, the cell tracks resulting from the simulation are formatted in the same format as the experimental data, and follow the same analysis pipeline. The code is available on Github.

## Acknowledgements

This study was funded through the SNSF Ambizione grant PZ00P3-161467 to GW and the SNSF grant 31003A-159673 to EvN.

## Additional information

### Funding

| Funder | Grant reference number | Author |
|---|---|---|
| Schweizerischer Nationalfonds zur Förderung der Wissenschaftlichen Forschung | PZ00P3_161467 | Guillaume Witz |
| Schweizerischer Nationalfonds zur Förderung der Wissenschaftlichen Forschung | 31003A_159673 | Erik van Nimwegen |

The funders had no role in study design, data collection and interpretation, or the decision to submit the work for publication.

### Author contributions

Guillaume Witz, Conceptualization, Resources, Data curation, Software, Formal analysis, Funding acquisition, Investigation, Visualization, Methodology, Writing—original draft, Project administration, Writing—review and editing; Erik van Nimwegen, Formal analysis, Funding acquisition, Methodology, Writing—review and editing; Thomas Julou, Resources, Formal analysis, Methodology, Writing—review and editing

## Author ORCIDs

Guillaume Witz https://orcid.org/0000-0003-1562-4265
Thomas Julou https://orcid.org/0000-0001-7123-198X

## Decision letter and Author response

Decision letter https://doi.org/10.7554/eLife.48063.sa1
Author response https://doi.org/10.7554/eLife.48063.sa2

# Additional files

## Supplementary files

• Transparent reporting form

## Data availability

Images of growth channels and MoMA segmentations have been deposited on Zenodo.

The following dataset was generated:

| Author(s) | Year | Dataset title | Dataset URL | Database and Identifier |
|---|---|---|---|---|
| Witz G, van Nimwegen E, Julou T | 2019 | Analysis of division and replication cycles in E. coli using time-lapse microscopy, microfluidics and the MoMA software | https://doi.org/10.5281/zenodo.3149097 | Zenodo, 10.5281/zenodo.3149097 |

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

**Appendix 1**

## Experiments statistics

**Appendix 1—table 1. Statistics for all experiments.** Glycerol auto is the dataset analyzed automatically, while Glycerol is the one analyzed manually. Each growth condition represent one experiment during which multiple positions on the chip where recorded and for which multiple growth channels were analyzed. The discarded fraction represents cell cycles not following exponential growth. In the automated analysis (Glycerol auto) an additional 14% of cycles are discarded because of a failed origin tracking. $r$ stands for Pearson correlation, and the $m-d$ superscript indicates a mother-daughter correlation. The doubling time ($1/\lambda$) is obtained by fitting the distribution of growth rates with a log-normal distribution.

| Experiment | Discarded % | # cell cycles | $1/\lambda\,[min]$ | Adder r | $\lambda^{m-d}$ r | $L_b^{m-d}$ r |
|---|---|---|---|---|---|---|
| Glycerol auto | 3.3 | 3070 | 86.0 | −0.10 | 0.33 | 0.45 |
| Glycerol | 2.1 | 810 | 89.0 | −0.07 | 0.42 | 0.58 |
| Glucose | 2.1 | 1035 | 53.0 | −0.04 | 0.47 | 0.66 |
| Glucose +AA | 2.4 | 1159 | 41.0 | −0.12 | 0.36 | 0.48 |

## Appendix 2

## Other models

In this article we have shown that models relying on the concept of initiation mass, as well as those involving a constant timer from initiation to division are incompatible with measurements. Still, those models are able to reproduce a wide range of experimental measurements, and we wanted to understand where they would break. We give here two examples of such an analysis. In the first case we tried to reproduce the model proposed in *Wallden et al. (2016)*. This model assumes that cells initiate replication around a specific initiation mass length $L_i$ and then grow for an amount of time depending on growth rate $T_{CD}(\mu)$ before dividing (*Appendix 2—figure 1A*). In panels B and C of *Appendix 2—figure 1* we show that we are successfully reproducing the model used for example in Figure 6 of *Wallden et al. (2016)*. The histogram of the number of origins at birth shown in *Appendix 2—figure 1D* shows a clear failure of the model a majority of cells in slow growth conditions are born with an ongoing round of replication in contradiction with experimental data (see e.g. Figure 3 of *Wallden et al., 2016*).

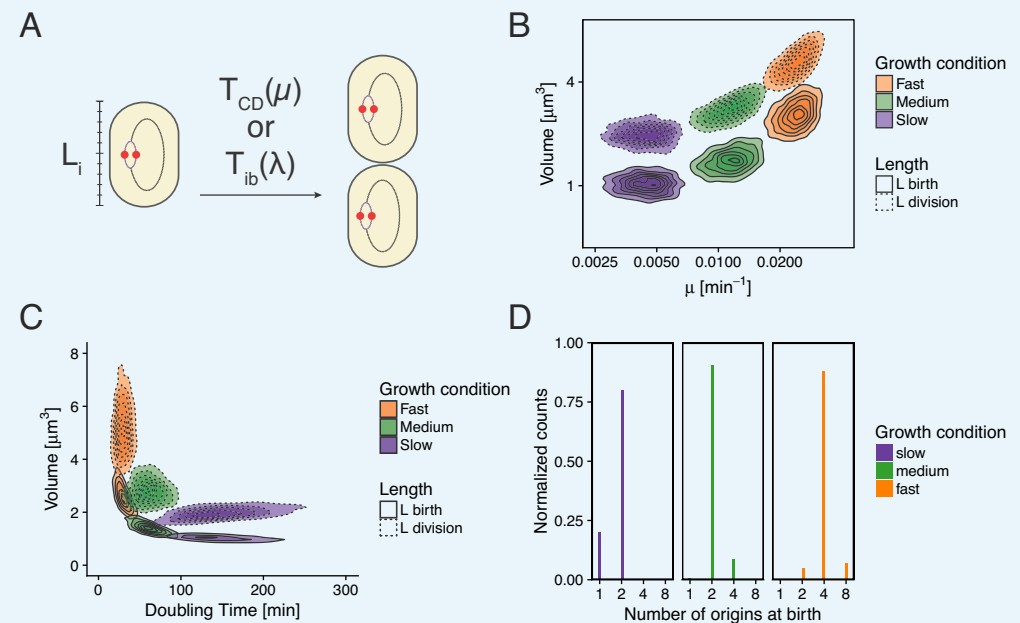

**Appendix 2—figure 1.** Re-implementation of the model proposed in *Wallden et al. (2016)* for three growth conditions. (**A**) Cells initiation at length $L_i$ and grow for a time $T_{CD}(\mu)$ before dividing. (**B**) Cell volume at birth and division as a function of growth rate. (**C**) Cell volume at birth and division as a function of generation time. (**D**) Distributions of the number of origins at birth.

The second model we are investigating here is based on the idea that replication and division are uncoupled and that the division cycle can be simply modeled as an adder (e.g. as in *Harris and Theriot, 2016*). As in our double-adder model, the replication cycle is controlled by an inter-initiation adder per origin. In very rare cases (0.4%), division happens in a cell with a single unreplicated origin. To avoid such rare unrealistic events, we condition division on the presence of two origins. The results are shown in *Appendix 2—figure 2*. The model surprisingly reproduces most of the features of the experimental data with one exception: the initiation to division variable $d\Lambda_{ib}$ is clearly not anymore an adder.

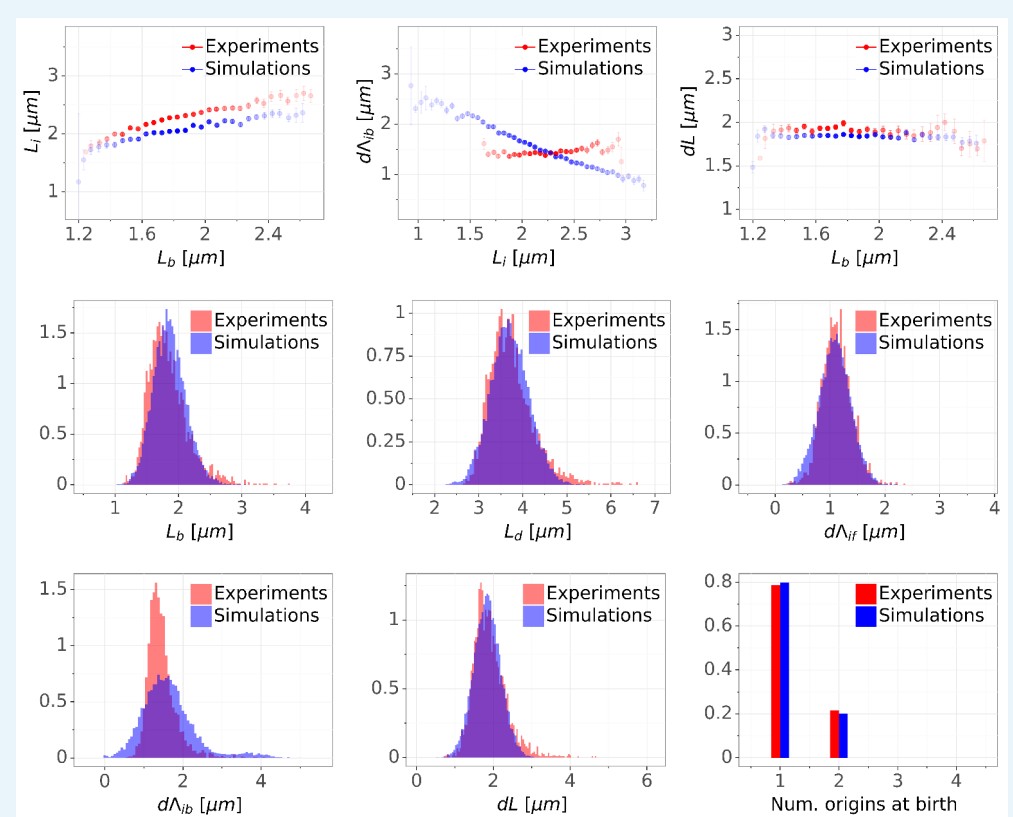

**Appendix 2—figure 2.** Comparison of distributions and correlations for slow growth case between experimental (M9+glycerol auto) and simulation data from a model combining an inter-initiation adder and a classic adder $dL = L_d - L_b$.

