## [Decision Letter]

**Acceptance summary:**

This paper uses time-lapse microscopy of individual cells to examine the coupling of chromosome replication and cell division. The authors present a new model suggesting a "dual adder" mechanism in which the volume added by cells controls both the subsequent round of replication initiation and cell division. This new model contrasts with recent single "adder" models. Although a dissection of the mechanistic basis remains to be done, this new model represents an important new concept that will guide future studies of the bacterial cell cycle.

**Decision letter after peer review:**

Thank you for submitting your article "Chromosome replication initiation controls both division and replication cycles in *E. coli* via a double-adder mechanism" for consideration by *eLife*. Your article has been reviewed by Gisela Storz as the Senior Editor, a Reviewing Editor, and three reviewers. The following individuals involved in review of your submission have agreed to reveal their identity: Alan Leonard (Reviewer #1).

The reviewers have discussed the reviews with one another and the Reviewing Editor has drafted this decision to help you prepare a revised submission.

Summary:

The regulatory mechanisms that coordinate the bacterial cell cycle remain obscure, but recent technological advances in single cell analysis have resulted in an avalanche of recent manuscripts focused on "adder mechanisms", particularly those related to cell size control where cells add constant length to their birth length and the added length fluctuates independently of either birth size or growth rate. While the "classic" adder model is insightful as a universal cell size maintenance mechanism, its application to other aspect of cell cycle regulation are less satisfying. The studies and model presented here in Witz et al., based on time lapse microscopy of individually growing cells, take the adder concept in an interesting new direction. The authors present an intriguing model that focuses on the initiation of chromosome replication as the critical starting point and incorporates "dual adder" mechanisms whereby cell volume increases from the initiation point controlling the subsequent cell division as well as the next initiation of replication. Importantly, the authors show that their model is compatible with the known parameters for both slow and fast growth conditions (Figure 5), parameters not usually examined or even addressed successfully in prior adder models. The authors also present a compelling comparison of their model with other models in Figure 6 and Figure 6—figure supplement 1. Despite the general enthusiasm for the work, the reviewers have several comments/concerns regarding the analyses presented as well as how the work relates to previous work. These issues will need to be addressed in a revision. Some of the following points are from different reviewers but hit on related points, so some revisions may address multiple points below.

Essential revisions:

1) The results of this study seem to disagree with those of Wallden et al. The latter does not find "adder" correlations in the slower conditions studied. The authors reference Wallden et al., but do not address the discrepancy.

2) The double-adder proposed do not capture the well-established exponential dependence of cell size on growth rate at the bulk level (Donachie, 1968). As such, their statement, in the Discussion section, on how models "fail to capture at least one important observation" may be applicable to the double-adder as well (at least in fast growth conditions).

3) The argument against the previous work of Wallden et al., in Appendix 2 seems incorrect. The authors state that "The histogram of the number of origins at birth shown in Figure 1D shows a clear failure of the model where cells in slow growth conditions are all born with an ongoing round of replication in contradiction with experimental data". But in fact, Figure 1D of this appendix is consistent with initiation occurring shortly after cell birth, which is precisely what the Wallden et al., experimental data shows in the slow growth condition. Moreover, the authors attempt to use (both in the main text Figure 5D and in Appendix 2) the bimodal distribution of the number of origin as a "smoking gun" for one model or another. But such bimodal distributions of origin numbers are mostly model-agnostic, and therefore cannot be used to discriminate between models.

4) In order to discriminate between models, a clearer statement of which models were considered is necessary. In particular, how is noise implemented exactly in the various models considered? e.g. does the noise affect the generation time or the growth rate, and how? What is the biologically dominant source of noise? One reason this might be important is that different implementations might affect the analysis used in Figure 6.

5) In Figure 6, the authors propose a new method for distinguishing between models. The authors should also test the method and validate it on synthetic data, to show that the method can indeed decouple the newly proposed model from previous ones. From the text and Figure 6, it is not obvious that correlation patterns similar to the double-adder cannot be generated by other models (e.g., can this method distinguish between the multiple origins accumulation model and the double-adder?).

6) The results of Figure 5C do not seem compelling. Could the authors perform some statistical analysis to test the data more quantitatively?

7) Regarding the Wallden model test (Appendix 2), it would be useful to show in panel D the data from Figure 3 of the Wallden paper, as to judge the claimed contradiction. Please then also provide a comparison with your model (Figure 5D), so that one can judge how well it predicts their data.

8) This also addresses another general topic: how similar are the data sets from the different studies? Is would be good to do this not only for 5D-style data, but also other quantifications. Are they qualitatively or quantitatively similar? If they are different, what are the causes?

9) In Figure 2 of the appendix, red and blue do not appear to be properly labelled.

10) Regarding the presentation of the modelling, I can guess what the 3x3 and the colouring means, but descriptions are not given in Appendix 3—figure 1, nor what the use of 'estimating independence' is. The former is given in Figure 6A, but no clarification is given about the aim of this exercise – e.g., why are these 4x4 the 'best decompositions' (how can I see that from these 4x4 diagrams)? But also, very basic aspects of their approach, like the notion that different rows in Figure 6A bottom are different models, is highly non-intuitive. In 6B caption, what is the difference between 'blue area', 'shaded blue'. What is the difference between the mathematical 'determinant of the correlation matrix' and the more usual correlation coefficient? Because of its unusual nature, more effort needs to go into explaining such basics. The authors did try to explain some aspects in the main text (subsection “The double-adder model best captures the correlation structure of the data”), but overall these quite crucial parts of the paper are very difficult to follow, even with a quantitative background.

11) The authors did a good job in discussing their results in relation to the possible molecular mechanisms, such as the possibility of replication initiation setting future division sites. This relates to recent work showing rapid min-system-driven repositioning of the FtsA rings in filamentous cells, which appear to be odds with such a notion (Wehrens, 2018), which would be good to discuss. On a different note, that study also found adder behaviour despite already having multiple chromosomes because of the filamented state, and single division events despite many potential division sites and rings, all suggesting some division to division regulation. It would be good to discuss these findings, also to keep open the possibility that such a division to division mechanism may exist, even if it is not required to explain the data presented here.

12) It would similarly be important to discuss surface to volume ratio models, and the extent to which they are consistent or not consistent with the presented data and models.

13) The authors do not discuss at the D period, or its compensations. I was trying to determine whether a birth-division adder has some bearing on this, but could not resolve it. The C period rather fixed in time, while the D period can compensate for (eg small) size at termination (Adicipingrum, 2015). That could be consistent with a birth-division adder, or at least both observations indicate the D period is not a constant time, as suggested in much of the literature, which would be good to discuss.

---

## [Author Response]

Essential revisions:1) The results of this study seem to disagree with those of Wallden et al. The latter does not find "adder" correlations in the slower conditions studied. The authors reference Wallden et al., but do not address the discrepancy.

There is indeed a discrepancy here. The same discrepancy has been already noted in (Si et al., 2019). In addition, (Si et al. 2019) re-analyzed the Wallden et al., data and found that these data in fact are more consistent with adder than sizer behavior. We clarify this point in the subsection "Cell size adder".

2) The double-adder proposed do not capture the well-established exponential dependence of cell size on growth rate at the bulk level (Donachie, 1968). As such, their statement, in the Discussion section, on how models "fail to capture at least one important observation" may be applicable to the double-adder as well (at least in fast growth conditions).

Our comment in the discussion pointed out that, for models other than our double-adder model, there is at least one experimental observation that is at odds with predictions of these models. This is not the case for our double-adder model, i.e. we are not aware that there are experimental observations that are at odds with this model. However, we do agree with the reviewer that some of the other models naturally predict that bulk average cell size should grow exponentially with growth rate, and that our double-adder model does not *require* such a relationship. However, our model is also not inconsistent with this relationship. In particular, if the initiation-to-division adder is implemented by the accumulation of a key molecule to a critical amount, and the rate of production of this molecule is independent of the bulk growth rate, then this will lead the average cell size to grow exponentially with bulk growth rate. We have added comments in the Discussion section to clarify these points.

3) The argument against the previous work of Wallden et al., in Appendix 2 seems incorrect. The authors state that "The histogram of the number of origins at birth shown in Figure 1D shows a clear failure of the model where cells in slow growth conditions are all born with an ongoing round of replication in contradiction with experimental data". But in fact, Figure 1D of this appendix is consistent with initiation occurring shortly after cell birth, which is precisely what the Wallden et al., experimental data shows in the slow growth condition. Moreover, the authors attempt to use (both in the main text Figure 5D and in Appendix 2) the bimodal distribution of the number of origin as a "smoking gun" for one model or another. But such bimodal distributions of origin numbers are mostly model-agnostic, and therefore cannot be used to discriminate between models.

We believe that the reviewer must have misunderstood what Figure 1D of the appendix and Figure 5D of the main text show because the distribution of the number of origins at birth does show a clear discrepancy between the predictions of the model of (Wallden et al., 2016) and experimental observations at slow growth. In Appendix 2—figure 1 we show results of our simulation of the model of (Wallden et al., 2016). As shown in panels B and C, our simulation successfully reproduces results of their model. However, as shown in Figure 1D, this model also predicts that in slow growth conditions, 80% of the cells should already have two origins *at birth.* That is, the model of (Wallden et al., 2016) predicts that in 80% of the cells, initiation has already taken place in the mother and the number of origins will go from 2 to 4 between birth and division. This is clearly at odds not only with our data at slow growth (Figure 5D) but also with the results of (Wallden et al., 2016) themselves which, as the reviewer correctly points out, show initiation from 1 to 2 origins taking place shortly after birth and that essentially no cells are observed that already have 2 origins at birth (e.g. see Figure S4 of (Wallden et al., 2016) and its caption "the typical number of origins at initiation are 1, 2, 4 for slow, intermediate and fast growth conditions"). We feel that it is important to point out this discrepancy between experimental observations and predictions of the model of (Wallden et al., 2016).

4) In order to discriminate between models, a clearer statement of which models were considered is necessary. In particular, how is noise implemented exactly in the various models considered? e.g. does the noise affect the generation time or the growth rate, and how? What is the biologically dominant source of noise? One reason this might be important is that different implementations might affect the analysis used in Figure 6.

There seems to be here a confusion between “different models" and “different decompositions". A similar confusion appears in other points raised by the referees, which made it clear to us that we insufficiently explained our approach. Crucially, different decompositions do not correspond to different models for generating data, but rather to different ways of parametrizing a dataset of cell cycles (which may either results from experimental measurements or from a simulation of a model). For example, the birth-to-division cycle of a cell can be fully characterized by a set of three variables. These might be directly measured variables, i.e. cell size at birth Lb, cell size at division Ld, and time between birth and division Tbd. But one can also use other sets of variables such as: size at birth Lb, added size dL=Ld-Lb and growth rate λ=log(Ld/Lb)/Tbd. We call such different sets of variables “decompositions". The basic idea behind the decomposition analysis of Figure 6 (now Figure 7) is that, if the cell-cycle control mechanisms operate on particular variables, the fluctuations in these variables should be independent of each other, i.e. the argument for an’ adder’ model of the birth-to-division cycle is precisely that added volume fluctuates independently of growth rate and size at birth, whereas other variables do not fluctuate independently. We thus devised an independence measure I that quantifies, for each possible decomposition, the extent to which the data supports that its variables fluctuate independently. Or to put it differently, if one were to implement a model in which the variables of the decomposition were chosen independently for each cell cycle, then I measures how close the correlation structure of such a model would be to the correlation structure of a given dataset. Importantly, Figure 7 does not compare results of simulations of different models but rather, compares different decompositions on two datasets: our experimental data from growth in glycerol, and results of simulations of our double-adder model. The latter is only included as a control to show that our independence measure I successfully identifies the set of variables that was used in the simulation. Since we clearly insufficiently explained the decomposition approach, we have completely reshaped subsection “The double-adder model best captures the correlation structure of the data”. We introduce it now in more detail and with the help of a new Figure 6 focusing on the simpler example of the division cycle (which was previously in Appendix 3) and have also simplified the mathematical explanation of independence measure I. We put the (unchanged) results of the full analysis in a new Figure 7. We hope that we clarified the description of the correlation matrices both in the text and in the legends. We also systematically added a color scale helping to understand the meaning of the colors in the matrices.

5) In Figure 6, the authors propose a new method for distinguishing between models. The authors should also test the method and validate it on synthetic data, to show that the method can indeed decouple the newly proposed model from previous ones. From the text and Figure 6, it is not obvious that correlation patterns similar to the double-adder cannot be generated by other models (e.g., can this method distinguish between the multiple origins accumulation model and the double-adder?).

The method is validated in Figure 7 by applying the decomposition analysis to simulated data of our double-adder model and showing that the top decomposition is indeed the set of variables used in the simulation and that other decompositions score much more poorly (by a large margin, i.e. the bottom rows of tables in Figure 7A). As highlighted in Figure 7—figure supplement 3, the decomposition that corresponds to the multiple origins accumulation model performs clearly worse both on experimental data (*I*=0.609, row 5) and on simulation data (*I*=0.487, row 11). The reason for this poor performance is easily explained by the fact that there is a strong correlation between the time Tib between initiation and the next division and the doubling time (see Figure 3A), which significantly lowers the independence score of this model. In order to verify more in-depth the validity of the decomposition method, we have now applied it on two additional datasets. As suggested by the reviewer, we created additional synthetic data by simulating an alternative model (Figure 7—figure supplement 4) and showed that the decomposition approach successfully recovers the variables used in that simulation. Importantly, we now also apply the method to an experimental dataset from another laboratory (Figure 7—figure supplement 5) of cells of a different strain growing in different growth medium and show that here too the decomposition method indicates that the double-adder model best describes the correlation structure in the data.

6) The results of Figure 5C do not seem compelling. Could the authors perform some statistical analysis to test the data more quantitatively?

There are several reasons why we do not expect the simulation to perfectly recover the distribution of cell sizes. For example, because all measurements are noisy, the distributions of the fluctuations that we infer from the measurements are a bit larger than they truly are, leading to slightly larger variation in the simulations than in the data. Our intent was to show that, even though the distribution of cell sizes is not used to set the parameters of the model, our model approximately correctly recovers their means and variances. In the revision we have clarified these points. First, we changed the panel and show now the means and variances of the distributions in a regular plot rather than the entire distributions. Second, we added a comment in the text about the reasons behind the limited accuracy of this aspect of our model (subsection “Simulations of the double-adder model”).

7) Regarding the Wallden model test (Appendix 2), it would be useful to show in panel D the data from Figure 3 of the Wallden paper, as to judge the claimed contradiction. Please then also provide a comparison with your model (Figure 5D), so that one can judge how well it predicts their data.

Unfortunately this is not possible as Figure 3 in (Wallden et al., 2016) only shows a distribution of replisomes as a function of cell size, from which it is not possible to extract initiation time (as stated by the authors “The detection of DnaQ does not allow us to reliably monitor the individual replication initiation events in individual cells"). However, it is clear that the data of (Wallden et al., 2016) are inconsistent with the prediction of their model that most cells already have 2 origins at birth (our Appendix 2—figure 1). First, from the distribution of initiation volumes as a function of birth volumes (Figure 4b where the authors used SeqA to monitor initiation) it is clear that the large majority of cells are initiating after birth, i.e. have only one origin at birth. Second, as already noted in point 3 above, the authors themselves state that in their slow growth conditions most cells are born with a single origin.

8) This also addresses another general topic: how similar are the data sets from the different studies? Is would be good to do this not only for 5D-style data, but also other quantifications. Are they qualitatively or quantitatively similar? If they are different, what are the causes?

As mentioned in point 5, we now applied our decomposition method to an additional dataset from the recent publication of (Si et al., 2019). This is to our knowledge the only other available large-scale dataset with initiation measurements in multiple consecutive cell cycles. For that dataset too, we find that the double-adder model best describes the data, showing the validity of our model across data from different studies. We added the decomposition analysis of the data of (Si et al. 2019) as Figure 7—figure supplement 5.

9) In Figure 2 of the appendix, red and blue do not appear to be properly labelled.

For the sake of clarity, we added a legend to all supplementary figures and made in particular sure Figure 2 is properly labelled.

10) Regarding the presentation of the modelling, I can guess what the 3x3 and the colouring means, but descriptions are not given in Appendix 3—figure 1, nor what the use of 'estimating independence' is. The former is given in Figure 6A, but no clarification is given about the aim of this exercise – e.g, why are these 4x4 the 'best decompositions' (how can I see that from these 4x4 diagrams)? But also, very basic aspects of their approach, like the notion that different rows in Figure 6A bottom are different models, is highly non-intuitive. In 6B caption, what is the difference between 'blue area', 'shaded blue'. What is the difference between the mathematical 'determinant of the correlation matrix' and the more usual correlation coefficient? Because of its unusual nature, more effort needs to go into explaining such basics. The authors did try to explain some aspects in the main text (subsection “The double-adder model best captures the correlation structure of the data”), but overall these quite crucial parts of the paper are very difficult to follow, even with a quantitative background.

As mentioned in point 4, we agree our presentation was not sufficiently clear and we completely rewrote the decomposition section, aiming specifically to clarify the questions that the reviewer raises. Please see point 4 for details.

11) The authors did a good job in discussing their results in relation to the possible molecular mechanisms, such as the possibility of replication initiation setting future division sites. This relates to recent work showing rapid min-system-driven repositioning of the FtsA rings in filamentous cells, which appear to be odds with such a notion (Wehrens, 2018), which would be good to discuss. On a different note, that study also found adder behaviour despite already having multiple chromosomes because of the filamented state, and single division events despite many potential division sites and rings, all suggesting some division to division regulation. It would be good to discuss these findings, also to keep open the possibility that such a division to division mechanism may exist, even if it is not required to explain the data presented here.

The work of (Wehrens et al., 2018) is indeed relevant for the discussion and we now cite and discuss it in the text. As this work does not provide data on replication or chromosome segregation, we find it hard to draw definite conclusions from these intriguing observations for models of cell cycle homeostasis during normal growth conditions. We cite and discuss this work in the Discussion section.

12) It would similarly be important to discuss surface to volume ratio models, and the extent to which they are consistent or not consistent with the presented data and models.

The Theriot model (Harris and Theriot, 2016) is primarily concerned with giving a mechanistic explanation to the adder behavior and does not try to model the replication cycle. Also, it assumes that the initiation and division cycles are uncoupled. In Appendix 2—figure 2 we show that such a model is at odds with our experimental observations. Also, some very recent work (Oldewurtel et al., 2019) suggests that some assumptions of that model are at odds with new observations. We now comment on the Theriot model in the article (subsection “Simulations of the double-adder model”). We also clarified that the model presented in Appendix 2—figure 2 in fact corresponds to the type of model that Theriot proposes, rather than the more refined class of models from (Micali et al., 2018).

13) The authors do not discuss at the D period, or its compensations. I was trying to determine whether a birth-division adder has some bearing on this, but could not resolve it. The C period rather fixed in time, while the D period can compensate for (e.g. small) size at termination (Adicipingrum, 2015). That could be consistent with a birth-division adder, or at least both observations indicate the D period is not a constant time, as suggested in much of the literature, which would be good to discuss.

While our data and modeling has clear implications for the total C+D period, our experimental setup does not allow us to directly measure termination and therefore we cannot decompose this period into the separate C and D periods. Also, our double-adder model does not make any assumptions about how the C+D period is composed out of separate C and D periods. Hence, we do not think that our data allow us to bring a constructive contribution on this specific question.